# Biological Characteristics and Osteogenic Differentiation of Ovine Bone Marrow Derived Mesenchymal Stem Cells Stimulated with FGF-2 and BMP-2

**DOI:** 10.3390/ijms21249726

**Published:** 2020-12-20

**Authors:** Sandra Gromolak, Agnieszka Krawczenko, Agnieszka Antończyk, Krzysztof Buczak, Zdzisław Kiełbowicz, Aleksandra Klimczak

**Affiliations:** 1Laboratory of Biology of Stem and Neoplastic Cells, Hirszfeld Institute of Immunology and Experimental Therapy, Polish Academy of Sciences, R. Weigla 12, 53-114 Wroclaw, Poland; sandra.gromolak@hirszfeld.pl (S.G.); agnieszka.krawczenko@hirszfeld.pl (A.K.); 2Department of Surgery, Faculty of Veterinary Medicine, Wroclaw University of Environmental and Life Sciences, pl. Grunwaldzki 51, 50-366 Wroclaw, Poland; agnieszka.antonczyk@upwr.edu.pl (A.A.); krzysztof.buczak@upwr.edu.pl (K.B.); zdzislaw.kielbowicz@upwr.edu.pl (Z.K.)

**Keywords:** bone marrow MSCs, osteogenic differentiation, bone repair, large animal model

## Abstract

Cell-based therapies using mesenchymal stem cells (MSCs) are a promising tool in bone tissue engineering. Bone regeneration with MSCs involves a series of molecular processes leading to the activation of the osteoinductive cascade supported by bioactive factors, including fibroblast growth factor-2 (FGF-2) and bone morphogenetic protein-2 (BMP-2). In this study, we examined the biological characteristics and osteogenic differentiation potential of sheep bone marrow MSCs (BM-MSCs) treated with 20 ng/mL of FGF-2 and 100 ng/mL BMP-2 in vitro. The biological properties of osteogenic-induced BM-MSCs were investigated by assessing their morphology, proliferation, phenotype, and cytokine secretory profile. The osteogenic differentiation was characterized by Alizarin Red S staining, immunofluorescent staining of osteocalcin and collagen type I, and expression levels of genetic markers of osteogenesis. The results demonstrated that BM-MSCs treated with FGF-2 and BMP-2 maintained their primary MSC properties and improved their osteogenic differentiation capacity, as confirmed by increased expression of osteocalcin and collagen type I and upregulation of osteogenic-related gene markers *BMP-2*, *Runx2*, *osterix*, *collagen type I*, *osteocalcin*, and *osteopontin*. Furthermore, sheep BM-MSCs produced a variety of bioactive factors involved in osteogenesis, and supplementation of the culture medium with FGF-2 and BMP-2 affected the secretome profile of the cells. The results suggest that sheep osteogenic-induced BM-MSCs may be used as a cellular therapy to study bone repair in the preclinical large animal model.

## 1. Introduction

Large bone defects and delayed fracture unions and non-unions, if not repaired effectively by the body, result in pain and lead to morbidity and prolonged, expensive hospitalization [1]. Moreover, bone disorders, such as osteoarthritis and osteoporosis, and the aging of the population constitute serious issues and challenging clinical problems. Thus, researchers are looking for new treatment modalities that will slow or prevent the development of these disorders [2]. Although orthopedic surgery has made great advances, the gold standard for bone defect repair is still dominated by autologous or allogeneic bone grafts. Nevertheless, clinical demands for bone grafts are far above the available amounts of traditional, natural bone auto- and allo-grafts, especially considering the impending global problem of obesity and aging. Moreover, there are limitations related to bone allografts, such as donor site complications, inferior healing compared to autologous grafts, and risk of disease and infective agent transmission [3].

Recent advances in cell-based therapies for regenerative medicine have supported the development of mesenchymal stem cells (MSCs) as an effective and minimally invasive alternative in bone repair [4]. MSCs are an attractive candidate for clinical approaches because of their ability to self-renew and differentiate into multiple tissues, including bone, cartilage, and fat [5]. They are a key component of the bone repair process, representing the precursors for bone-forming osteoblasts and cartilage-forming chondrocytes and modulating the healing response [4]. The cellular and molecular signals of a bone defect correspond to the beginning stages of fracture healing, which involves, inter alia, the trafficking and activation of MSCs. Both innate and adaptive immune responses are modulated by MSCs. Therefore, MSCs not only provide progenitor cells, but also activate other cells involved in tissue regeneration via the paracrine effect, thereby modulating a favorable microenvironment and coordinating bone remodeling [6]. Molecular osteoinductive events in bone regeneration involve a variety of growth factors, cytokines, and chemokine signaling pathways, as well as the upregulation of genes involved in bone formation and mineralization [4,7].

Effective cellular therapy for bone regeneration employs MSCs and growth factors that enhance different steps of differentiation. Bone morphogenetic protein-2 (BMP-2) and fibroblast growth factor-2 (FGF-2) are two molecules that have a synergistic osteogenic effect on MSCs [7,8,9,10]. FGF-2 regulates the migration, proliferation, and differentiation of many cell types, including vascular endothelial cells and osteoblasts. There are studies demonstrating the potential of FGF-2 in promoting bone formation and angiogenesis [11,12,13,14,15]. Moreover, FGF-2 increases the BMP-2 osteoinductive potential by upregulating BMP-2 and its receptor expression level [10,16]. BMP-2 is a subtype of the BMP family of growth factors, which are mainly synthetized and secreted by osteoblasts [17]. BMP-2 regulates osteogenesis and induces MSCs to differentiate into cartilage and bone [18,19]. Furthermore, the United States Food and Drug Administration has approved the recombinant human bone morphogenetic protein-2 (rhBMP-2) for clinical applications [20,21]. However, the FGF-2/BMP-2 synergistic effect on bone regeneration is not fully explained, and requires further examination before safe clinical application can be achieved.

In the present study, we introduce the synergistic effect of FGF-2 and BMP-2 on the pro-osteogenic potential of bone marrow-derived mesenchymal stem cells (BM-MSCs) derived from sheep. Whereas rodents are the most widely used animal models for translational medicine, preclinical large animal models provide a better understanding of the mechanism of human diseases. For instance, when investigating musculoskeletal diseases, the thin cartilage and bones of rodents represent an inadequate volume and size of the defects, and the lack of the ability to manage long-term studies makes them less useful for preclinical studies than large animal models [22,23]. Therefore, we decided to use ovine BM-MSCs to study the effect of FGF-2 and BMP-2 on MSC proliferation, phenotype, osteogenesis-related proteins (osteocalcin and collagen type I) expression, multilineage differentiation, and, in particular, early and late osteogenesis-related gene marker expression. We also evaluated the cytokine expression level of BM-MSCs treated with FGF-2, with and without BMP-2. The comprehensive characteristics of ovine BM-MSCs make them good candidates for the study of bone reconstruction in veterinary medicine using a large animal model as a preclinical model in order to assess of the efficacy of cellular therapy in bone regeneration for potential application in clinical practice.

## 2. Results

### 2.1. Morphology of Sheep BM-MSCs

The ovine bone marrow-derived mesenchymal stem cells altered their morphology when cultured for 21 days in: (1) complete αMEM, (2) αMEM supplemented with FGF-2, (3) αMEM supplemented with FGF-2 and BMP-2, (4) osteogenic differentiation medium, (5) osteogenic differentiation medium supplemented with FGF-2, and (6) osteogenic differentiation medium supplemented with FGF-2 and BMP-2. In the αMEM-cultured medium, deprived of additional cytokines, BM-MSCs showed a spindle-shaped morphology, typical for MSCs (Figure 1a). Likewise, in an αMEM medium supplemented with FGF-2, the cells were spindle-shaped, although they were smaller and denser (Figure 1c). However, the largest morphological differences were observed in the cell culture treated with both BMP-2 and FGF-2, where some cells grew into fine mesh-like structures, and some covered the cell monolayer rounded aggregates, resembling bone-like structures (Figure 1e). When BM-MSCs were cultured in an osteogenic differentiation medium, they exhibited calcium deposition, regardless of whether they were additionally stimulated with BMP-2 and/or FGF-2 (Figure 1b,d,f). Interestingly, the crystal formation in the osteogenic differentiation medium supplemented with BMP-2 and/or FGF-2 tended to be smaller, denser, and more dispersed throughout the monolayer (Figure 1d,f).

### 2.2. Cell Proliferation and Doubling Time Analyzed with the MTT Assay

The proliferative activity of sheep BM-MSCs (passage 3) cultured in complete αMEM (control) and αMEM supplemented with BMP-2 and/or FGF-2 was evaluated using the MTT assay. Growth curves in the cells in all culture conditions appeared as a typical “S” curve and showed a latency phase of about 4 h after seeding. After 24 h of incubation, the cells proliferated rapidly and entered the logarithmic phase. After approximately 72 h, the BM-MSCs entered the plateau phase and began to degenerate (Figure 2a). Compared to the untreated cells (cultured in αMEM), the proliferation of the cells treated with BMP-2 and/or FGF-2 began to increase after 24 h, and this stimulatory effect increased during observation. The most efficient proliferative activity was observed in the cells treated with FGF-2, compared to control (*p* < 0.005).

The doubling time of ovine BM-MSCs differed depending on cell culture conditions (Figure 2b). BM-MSCs doubled their cell number significantly faster when cultured in a medium supplemented with BMP-2 and/or FGF-2, compared to the control αMEM without any additional cytokines. BM-MSCs treated with FGF-2 had the strongest cell proliferative ability.

### 2.3. Immunophenotype of Sheep BM-MSCs in Different Culture Conditions

BM-MSCs from sheep bone marrow (passage 3) expressed the specific MSC surface markers CD73 and CD105 when cultured in the control medium αMEM and a medium supplemented with BMP-2 and/or FGF-2 on days 7 and 14 (Appendix A) and day 21 (Figure 3). The values of positive population ranged between 84% and 99%. The values of CD73 and CD105 for BM-MSCs cultured in αMEM with or without FGF-2 were at the same high level of above 91% of the population, regardless of culture time. On day 7, BM-MSCs cultured with FGF-2 and BMP-2 showed a lower level of CD73 and CD105 positive cells (84–87%), compared to cells treated with FGF-2 alone or without any supplements (Appendix A). The percentage of CD73-positive BM-MSCs increased over time for all cell populations cultured in three different conditions. The level of cells with the hematopoietic phenotype CD34 was very low, 1–6% of positive cells, among which the highest level of 5–6% was reported for the cells cultured for 14 days in αMEM with FGF-2 and BMP-2 and for the cells incubated for 21 days in the control medium αMEM. As with CD34, the common hematopoietic marker CD45 was also at a low level between 0–8% during observation. The major histocompatibility complex class II antigen HLA DR was detected at the beginning of the BM-MSCs culture (7 days) in the media supplemented with BMP-2 and/or FGF-2, with values ranging between 38% and 45% of the population, whereas in the control medium αMEM, the value was assessed at 4.5%. The level of HLA-DR-positive cells decreased over time, and on day 21 of culture, was assessed at below 1% in all culture conditions (Appendix A).

### 2.4. CD90 Expression Level in Sheep BM-MSCs

As there are no commercially available anti-CD90 antibodies reactive with ovine cells for flow cytometry, CD90 was assessed using reverse transcription polymerase chain reaction (RT-PCR). Differences in the *CD90/GAPDH* relative fold change level depended on the culture medium conditions. Gene expression for the MSC marker *CD90* was detected in all cells treated with or without BMP-2 and/or FGF-2. However, *CD90* expression was associated with culture medium supplementation, and the highest level of *CD90* was observed for BM-MSCs cultured in the complete αMEM medium, whereas in the BM-MSCs treated with FGF-2 and BMP-2, *CD90* relative fold change was at the lowest level (6.51 vs. 2.95 on day 7; *p* < 0.005). The *CD90* expression remained at a similar level within the cells in the same culture conditions, regardless of incubation time (Figure 4).

### 2.5. Assessment of Osteogenic Differentiation Markers

To assess the effect of FGF-2 and BMP-2 on the expression of osteogenesis-related proteins, immunofluorescent staining of osteocalcin and collagen type I was performed after 21 days of BM-MSC culture in an osteogenic differentiation medium as a control and an osteogenic differentiation medium supplemented with BMP-2 and/or FGF-2. The cells treated with BMP-2 and/or FGF-2 showed a moderately enhanced expression of both osteogenic differentiation markers compared to the control cells (Figure 5a,b). The immunofluorescence intensity of osteocalcin and collagen type I was higher in the cells treated with both BMP-2 and FGF-2 than in the cells treated with FGF-2 alone (Figure 5c,d).

### 2.6. Effect of BMP-2 and FGF-2 on mRNA Expression in Genes Involved in Osteogenic Differentiation

The gene expression of the early osteogenic differentiation markers: *bone morphogenetic protein-2* (*BMP-2*), *runt-related transcription factor 2* (*Runx2*), *osterix* (*Osx*), and *collagen type I* (*ColI*), and late osteogenic markers: *osteocalcin* (*Ocl*) and *osteopontin* (*Opn*), following 7, 14 and 21 days of incubation in the αMEM medium with BMP-2 and/or FGF-2, was analyzed as the effect of BMP-2 on sheep BM-MSCs in osteogenesis. The expression of *BMP-2* increased after 14 days of BM-MSC incubation in both cell culture conditions, compared to control. However, there were no significant differences in the expression level of this gene between 14 and 21 days of cell culture. BM-MSCs cultured 14 days in the medium with FGF-2 and BMP-2 exhibited a higher expression of *BMP-2* than those cultured in the medium with only FGF-2 (Figure 6a, RQ 3.97 vs. 2.34; *p* ˂ 0.0001). The mRNA expression of *Runx2* increased over time from day 7 to 14 in the BM-MSCs cultured in the medium supplemented with FGF-2 and BMP-2 (Figure 6b, RQ 1.84 vs. 6.78; *p* ˂ 0.0001), whereas in the cells cultured only with FGF-2, the expression level of *Runx2* was downregulated over time, but still higher than control. The highest peak in *Osx* gene expression was observed on day 14 in the BM-MSCs treated with FGF-2 and BMP-2 (Figure 6c, RQ 3.33; *p* ˂ 0.0001); afterwards, on day 21, gene expression decreased. In the BM-MSCs treated only with FGF-2, the *Osx* expression level increased after 14 days of incubation and then decreased. In the control, *Osx* gene expression increased over time, and on day 21, it was even slightly higher than in FGF-2-treated cells. *ColI* gene expression was upregulated in the BM-MSCs treated with FGF-2 or FGF-2 together with BMP-2 over time, and the highest expression level was reported after 21 days of incubation (Figure 6d RQ 5.19 and 4.25; *p* ˂ 0.0001). Interestingly, *ColI* expression was also upregulated in the control on day 14, but decreased afterwards. The gene expression of the late osteogenic marker *Ocl* increased over time for both BM-MSCs treated with FGF-2 alone or FGF-2 with BMP-2; however, at the time point of 21 days, it was higher in the cells incubated with both FGF-2 and BMP-2 (Figure 6e, RQ 7.20 vs. 4.81; *p* ˂ 0.0001). The effect of FGF-2 and BMP-2 on the expression level of the second late osteogenic marker *Opn* also differed from that for FGF-2-only supplementation without BMP-2. *Opn* expression increased over time in both stimulated BM-MSCs. Nevertheless, the BM-MSCs cultured in the medium with FGF-2 and BMP-2 exhibited a significantly higher expression level than those cultured only with FGF-2 (Figure 6f, RQ 4.18 vs. 2.59; *p* ˂ 0.0001).

### 2.7. Differentiation Potential of Sheep BM-MSCs

The capacity of sheep BM-MSCs for multilineage differentiation was confirmed using Alizarin Red S staining for osteogenesis, Oil Red O staining for adipogenesis, and Alcian Blue staining for chondrogenesis. The impact of BMP-2 and FGF-2 on differentiation capacity was also assessed. The cells that were cultured for 21 days in an osteogenic differentiation medium and an osteogenic differentiation medium supplemented with BMP-2 and/or FGF-2 formed a mineralized matrix. The addition of the cytokines FGF-2 or FGF-2 together with BMP-2 did not have a significant impact on the osteogenic differentiation potential. BM-MSCs themselves had the most powerful osteogenesis capacity in comparison to adipogenesis and chondrogenesis; therefore, stimulation by cytokines did not change their osteogenic potential. Interestingly, BM-MSCs differentiate more efficiently into chondrocytes when treated with FGF-2 and/or BMP-2 (Figure 7).

### 2.8. Secretion Profile of Sheep BM-MSCs

The expression levels of 18 ovine cytokines in the supernatants from BM-MSCs cultured in different media: (1) αMEM, (2) αMEM supplemented with FGF-2, and (3) αMEM supplemented with FGF-2 and BMP-2, were analyzed using the semi-quantitative C-Series Ovine (Sheep) Cytokine Array C1 Kit to determine the impact of culture media on their secretion profile (Figure 8a). Furthermore, changes over time in cytokine expression were also examined, following treatment with BMP-2 and/or FGF-2 for 7, 14 and 21 days (Figure 8b,c). A comparison of cytokine profiles depending on the cell culture medium showed that in all culture conditions, decorin, the cytokine that influences fibrillogenesis, was at the highest relative expression level, which is more than 100% of the internal positive control. In the supernatant of the control cells cultured in the complete αMEM medium, eight cytokines had an expression level of ≥10%, among which the most abundant were immunomodulatory and pro-angiogenic cytokines, such as: decorin (>100%), regulated on activation, normal T-cell expressed and secreted–RANTES (~73%), interleukin-8–IL-8 (~42%), monokine induced by gamma interferon–MIG (~21%), allograft inflammatory factor–AIF (~17%), tumor necrosis factor α–TNF-α (~12%), secreted frizzled-related protein 3–sFRP-3 (~12%), and vascular endothelial growth factor A–VEGF-A (~11%). After the FGF-2 treatment, the expression levels of MIG, sFRP-3, TNF-α, and VEGF-A increased, whereas the expression level of IL-8 and RANTES decreased. The addition of FGF-2 and BMP-2 to the culture medium reduced the expression level of IL-8, TNF-α, and VEGF-A (Figure 8a).

Interestingly, the cytokine secretion profile of BM-MSC culture in the treated cells changed over time. In the supernatants from the cell culture treated with FGF-2, the expression level of RANTES increased gradually over time. The expression level of other cytokines remained at the same level in the supernatants collected after 7 and 21 days; however, after 14 days, the expression of most of the examined cytokines was the lowest (Figure 8b). The expression level of AIF, decorin, IL-8, sFRP-3 decreased over time in the supernatants from BM-MSCs treated with FGF-2 and BMP-2. The expression level of RANTES in the collected supernatant was the highest after 14 days (Figure 8c).

## 3. Discussion

MSC-based therapies are gaining popularity as a therapeutic tool in bone pathologies [24] thanks to their properties of self-renewal, differentiation into osteogenic lineage, secretion of a variety of biological factors, and the ability to regenerate damaged tissues. The application of MSCs in bone engineering can serve as an alternative when standard clinical methods are insufficient, especially for restoring large bone defects or difficult-to-treat non-union fractures. For a safe and efficient bone-healing therapy involving MSCs, it is necessary to characterize in detail the biological properties of MSCs and standardize cell culture conditions [25]. To enable the transition of MSCs from the laboratory to clinical use, large animal models (including sheep) have been widely investigated due to their structural and physiological similarities to humans. Before clinical application, cellular and molecular interactions are studied in vitro. Afterwards, small animal models facilitate the initial “proof of concept,” whereas large animal models allow for the examination of the safety, dosing, and efficacy of MSCs prior to clinical application [26]. Although human MSCs have been extensively investigated, sheep MSCs are still poorly characterized [27]. Another problem in research on MSC application in bone diseases is the efficacy of osteogenic differentiation. Consequently, it is crucial to investigate the factors that induce the osteogenic differentiation capacity of MSCs [24]. Previous studies report that bone formation consists of two phases, proliferation and mineralization, which are regulated by specific gene expression [28,29]. In this study, ovine bone marrow-derived MSCs, treated with FGF-2 as a cell proliferation stimulator and BMP-2 as an osteogenic inductor, were investigated in the context of their biological activity for a potential application as a cell-based therapy supporting bone healing in the large animal model.

Our results revealed that the morphology and proliferation of ovine MSCs isolated from bone marrow depended on culture medium supplements, which was previously reported for some supplements (FGF-2, ascorbic acid, epidermal growth factor, and platelet-derived growth factor-BB) in human MSCs [30] and ovine MSCs [31]. However, it has not yet been investigated how FGF-2 and BMP-2 affect the biological properties of ovine MSCs. Major changes in morphology were observed in BM-MSCs cultured in αMEM supplemented with both FGF-2 and BMP-2. In comparison to the morphology of the BM-MSCs cultured in the control medium, which showed a typically fibroblast-like spindle shape, in the cell culture treated with FGF-2 and BMP-2, some osteoblast-like structures were visible in addition to the standard fibroblastic morphology. This observation confirmed the pro-osteogenic function of BMP-2 when applied together with FGF-2 in a cell culture medium, and this phenomenon was also shown earlier for human and mouse MSCs [7]. Supplementation with FGF-2 alone did not result in new osteoblast-like structures; nevertheless, the cells were smaller and grown in a higher cell density. The effect of the osteogenic induction of BMP-2 and FGF-2 was also reported when the cells were cultured in a supplemented osteogenic differentiation medium [7]. The density of calcium deposits and mineralized matrix in the BMP-2- and/or FGF-2-induced cells was moderately higher compared to the cells cultured in an untreated osteogenic differentiation medium. A comparison of cell proliferation curves according to stimuli showed that the highest proliferation rate occurred in αMEM supplemented with FGF-2. This observation is in line with previous reports documenting that FGF-2 increased the proliferation of MSCs and other cell types [32,33,34,35]. The addition of both cytokines, FGF-2 and BMP-2, to the culture medium decreased the proliferation of BM-MSCs, although it was still higher than in the cells cultured in the control αMEM medium. This phenomenon, with high probability, is associated with an advanced osteogenic differentiation potential triggered by the complementary action of FGF-2 and BMP-2.

To fulfill the minimal criteria of the International Society for Cellular Therapy (ISCT) relating to the phenotype and tri-lineage differentiation characteristics of MSCs [36], ovine BM-MSCs were characterized for their immunophenotype and multi-lineage differentiation potential. The expression profile of the surface antigens of ovine MSCs has already been reported in some studies [30,31,37,38,39]; however, there is a lack of data about the effect of FGF-2 and BMP-2 on ovine BM-MSC surface markers. Bearing in mind the restrictions concerning the critical quality of the MSCs that are under consideration for clinical application, it is crucial to analyze whether FGF-2 and BMP-2, which have a positive effect on osteogenic differentiation, do not have a negative effect on the biological properties of MSCs over a period of differentiation time. Therefore, in our study, BM-MSCs treated with osteogenic stimuli were examined during the follow-up period up to 21 days of the cell culture for CD73, CD105, CD34, CD45, and HLA DR expression levels using flow cytometry and for CD90 using RT-PCR. The analysis revealed a high expression level of CD73, CD90, and CD105, and a very low or lack of the expression of CD34, CD45, and HLA DR; this confirmed that the ovine BM-MSCs maintained the naïve MSC phenotype according to ISCT criteria [36]. Moreover, BM-MSCs stimulated with BMP-2 and/or FGF-2 maintained the stability of the basic MSC phenotype, and the level of CD73 and CD105 positive cells increased during observation time, thus confirming the purity of the BM-MSC population. However, our data showed an alteration in the CD90 gene expression performed using the RT-PCR method, depending on medium supplements and observation time. The highest CD90 gene expression was observed in the control cells cultured in αMEM without any cytokine addition, and the lowest expression was observed in BM-MSCs cultured in αMEM supplemented with FGF-2 and BMP-2 after 21 days of cultivation. This observation suggests that the decrease of CD90 gene expression is a result of FGF-2 and BMP-2 activity. This phenomenon may be the effect of an ongoing osteogenic differentiation of the examined ovine BM-MSCs, as demonstrated in studies on the osteogenic differentiation of human dental pulp stem cells and a decreased expression of the MSC surface marker CD90 [40]. The data showed little variation in the expression level of the negative MSC surface markers, especially HLA DR. Initial analysis, performed on day 7, revealed that the level of cells with the expression of the HLA DR antigen constituted almost half of the population of ovine BM-MSCs treated with BMP-2 and/or FGF-2. However, HLA DR expression decreased to below 0.3% in all examined populations after 21 days of cultivation, which proved that over time, cells with a potential activation ability were removed from the ovine BM-MSC population. Nevertheless, the variability in the expression of HLA DR for human BM-MSCs used in clinical studies has been already described by Grau-Vorster et al. They reported that the multilineage differentiation and immunomodulatory potential of MSCs was independent of their HLA DR expression level [41]. Our results concerning the immunophenotype suggest that BM-MSCs treated with BMP-2 and/or FGF-2 in long-term culture did not lose their characteristic MSC properties.

The FGF-2 and BMP-2 treatment affected osteocalcin and collagen type I expression, as indicated by immunofluorescence intensity. Ovine BM-MSCs showed a moderately high expression of osteocalcin and collagen type I when cultured in the osteogenic differentiation medium supplemented with both FGF-2 and BMP-2. Nevertheless, the addition of FGF-2 alone also slightly increased the expression of osteocalcin and collagen type I at the protein level compared to the osteogenic differentiation medium without any supplements. This is a significant observation that confirms the pro-osteogenic stimulatory effect of FGF-2 and BMP-2 on ovine BM-MSCs. Nevertheless, the differences in immunofluorescence intensity between the cells cultured in an osteogenic medium with or without supplements were rather modest. The reason may have been the impact of the osteogenic differentiation medium on MSCs, which significantly induce cells to differentiate into the osteogenic lineage, regardless of supplement addition. The role of FGF-2 is to enhance the osteogenic potential by stimulating cell proliferation during the early stages of osteogenesis and regulating the BMP-2 osteoinductive potential, whereas BMP-2 is known as one of the strongest inducer of osteogenesis. In physiological conditions, FGF-2 and BMP-2 are produced by osteoblasts and accumulated in the extracellular matrix of the bone [7]. In our study, both cytokines added to αMEM were able to promote the osteogenic differentiation of MSCs by inducing the expression of a range of osteogenic gene markers.

The degree of sheep BM-MSC osteogenic differentiation following FGF-2 and BMP-2 treatment was determined by assessing the expression level of the osteogenic marker genes: *BMP-2*, *Runx2*, *osterix*, *collagen type I*, *osteocalcin*, and *osteopontin*. Our study demonstrated that FGF-2 and BMP-2 treatment upregulated the expression of osteogenic signaling molecules in the sheep BM-MSCs, starting from the early steps of bone formation to the late phase of mineralization. The peak expression of *BMP-2* in BM-MSCs cultured with FGF-2 and BMP-2 was observed after 14 days of differentiation and was higher than in the cells cultured without BMP-2. *BMP-2* has already been described by other researchers as an important regulator of osteogenic differentiation [42,43]. The finding that BMP-2 increased its own expression may suggest that during the bone healing process, BMP-2 induces its own expression in response to injury, making it a feedback regulation. It is worth noting that FGF-2 alone also enhanced the relative *BMP-2* expression, but the difference in its expression between days 7, 14, and 21 was rather low.

*Runx2* has been reported as the earliest osteogenic marker and a stimulator of *osterix* expression [44,45]. Both genes are involved in osteogenesis as master transcription factors. *Runx2* may affect the early stage of the recruitment of osteoblastic progenitor cells, and *osterix* is involved in the final osteogenic differentiation stage. Furthermore, an overexpression of *osterix* induces the expression of the final osteogenic marker, *osteocalcin* [46]. These observations support the results of the present study. The relative expression level of *Runx2* increased after 14 days of incubation and was the highest in BM-MSCs treated with FGF-2 and BMP-2, which confirmed the stimulatory effect of FGFs and BMP-2 on *Runx2* expression, as demonstrated by other researchers [19,47]. However, BMP-2 significantly upregulated the *osterix* gene expression on day 14 of culture. In contrast, FGF-2 downregulated the relative *osterix* gene expression in ovine BM-MSCs over the culture period. The same outcome in the gene expression of the late osteogenic markers, *osteocalcin* and *osteopontin*, was observed in BM-MSCs stimulated with FGF-2 and BMP-2. The expression level of these genes increased over time and was the highest after 21 days of incubation. Similar results for *Runx2*, *osterix*, and *osteocalcin* in human BM-MSCs treated with BMP-2 have already been reported [19]. However, in this study, the osteogenic differentiation medium supplemented with BMP-2 was used instead of the complete αMEM.

FGF-2 and BMP-2 are involved in different stages of bone repair. FGF-2 enhances bone formation at the early stage of differentiation based on the proliferation phase, whereas BMP-2 plays a key role in the final stages of mineralization [48]. This hypothesis is supported by the results of our work relating to *collagen type I* and *osteopontin* expression. Collagen is one of the main proteins synthesized by osteoblasts in the early phase of osteogenic differentiation [49]. Relative *collagen type I* gene expression was the highest in the sheep BM-MSCs treated with FGF-2 on day 21. Interestingly, BMP-2, as an active inducer of osteogenic differentiation, enhanced the expression of the late osteogenic marker, *osteopontin*. These findings confirmed that FGF-2 and BMP-2 affect bone formation at different stages of osteogenic differentiation.

To identify the role of BMP-2 and FGF-2 in MSC differentiation, sheep BM-MSCs were cultured in commercial osteo-, adipo-, and chondrogenic differentiation mediums with or without BMP-2 and/or FGF-2. This study showed that BMP-2 and/or FGF-2 did not significantly affect the Alizarin Red S staining intensity. This finding can be explained by the fact that ovine BM-MSCs have a good ability to differentiate into osteoblasts when cultured in an osteogenic differentiation medium. Therefore, the differences between Alizarin Red S stained cells were rather difficult to distinguish, as the intensity of staining in all treated or untreated cells was very high. However, FGF-2 and BMP-2 treatment improved the efficacy of chondrogenic differentiation. The stimulatory effect of BMP-2 on the chondrogenesis of human MSCs has already been reported by other authors [50,51]. Our results confirmed a similar effect of BMP-2 on MSCs obtained from sheep.

MSCs are known for their capability to produce many growth factors, cytokines, and chemokines, which affect immune cells and types of cells that modulate the local environment during regeneration [52]. In this study, 18 cytokines were screened on sheep BM-MSCs following FGF-2 and/or BMP-2 treatment. The cytokine expression pattern differed depending on the cell culture conditions and cultivation time. Firstly, it is worth noting that in all supernatants, the level of decorin was significantly high. Decorin is a bone matrix protein that plays a key role in bone remodeling [53]. Amable and her team have also reported that human MSCs derived from bone marrow and adipose tissue secrete decorin [54]. Moreover, Shu et al. have found that FGF-2 and FGF-18 upregulate the decorin gene expression level in BM-MSCs [55]. Their results support our observation that FGF-2 exerts a stimulatory effect on decorin secretion by ovine BM-MSCs. Our data also showed that the level of proangiogenic and immunomodulatory cytokines, namely VEGF-A, MIG, sFRP-3, and TNF-α, increased in FGF-2-treated BM-MSCs. This observation is in line with a study performed by Gorin et al., who found that FGF-2 enhances the angiogenic properties of the human dental pulp-origin MSC secretome [56]. In contrast, stimulation with both FGF-2 and BMP-2 decreased VEGF-A and TNF-α. Interestingly, IL-8 was downregulated in cells treated with both cytokines, FGF-2 and BMP-2, and its highest level was observed in the control culture without any stimuli. IL-8 is an inflammatory mediator that plays an important role as an enhancer of cell migration in tissue repair. However, Bastidas-Coral et al. demonstrated that IL-8 did not stimulate the osteogenic differentiation of human MSCs [57]. Based on these findings and our observations, we can hypothesize that the upregulation of IL-8 is not needed for osteogenesis.

The role of FGF-2 combined with BMP-2 in the osteo-induction of MSCs and osteoblast-like cells has been investigated by different studies [7,16,48]. For example, it was found that the co-delivery of low doses of FGF-2 and BMP-2 is more efficient in the bone formation of old human and mouse cell cultures than BMP-2 treatment alone [7]. Although the effect of stimulation with FGF-2 and BMP-2 on the osteogenic differentiation of mouse and human MSCs remains to be researched, the primary MSCs obtained from large animals have not yet been investigated for their osteogenic potential following FGF-2 and BMP-2 treatment. It is important to fully understand the response to osteogenic stimuli of ovine cells in vitro before undertaking in vivo experiments on the large animal model in order to test different strategies of enhancing the regeneration of large bone defects. Our results suggest that treatment with FGF-2 and BMP-2 significantly improves the osteogenic potential of ovine BM-MSCs on both the molecular and the protein level.

## 4. Materials and Methods

### 4.1. BM-MSC Isolation and Culture

The study was approved by the institutional Animal Ethics Committee at the Institute of Immunology and Experimental Therapy PAS (No 63/2017 from 21 of June 2017). Six adult sheep weighing from 42 to 54 kg were used in the study. A 15-guage bone marrow aspiration needle (15 G × 15 mm, Perfectus, Medax, Italy) was used to aspirate 10 mL of bone marrow from the iliac crest into a 20 mL heparinized syringe. All procedures were performed under general inhalant anesthesia with oxygen volatilized isoflurane (IsoVet, Piramal Healthcare, UK). Analgesia was provided with fentanyl (Fentanyl WZF, Warsaw, Poland). Bone marrow aspirations were performed by experienced surgeons at the Department and Clinic of Surgery, Faculty of Veterinary Medicine, Wroclaw University of Environmental and Life Sciences. Mononuclear cells were isolated using the Lymphoflot gradient (Bio-Rad, Dreieich, Germany, cat. no. 824012) during 30 min of centrifugation at 400 g at room temperature (RT). Next, the isolated mononuclear cells of bone marrow-origin were collected and washed two times in PBS. Cell suspensions were seeded in 75 cm^2^ (T75) culture flasks and cultured in Minimum Essential Medium α-transformation–αMEM (IIET PAS, Wroclaw, Poland) supplemented with 10% fetal bovine serum, FBS (Biowest, Riverside, Montana, MT, USA, cat. no. S1810-500), 2 mM L-glutamine (Biowest, Riverside, Montana, MT, USA, cat. no. X0550-100), and 1% penicillin/streptomycin (Merck, Saint Louis, MO, USA, cat. no. P0781). The cells were incubated in a humidified atmosphere at 37 °C with a 5% CO_2_. To ensure an effective adherence of MSCs, the culture medium was first changed after 7 days of incubation, after which the complete medium was replaced every three days. When the plastic adherent bone marrow-derived mesenchymal stem cells (BM-MSCs) reached 80% confluence, they were harvested using the Accutase Cell Detachment Solution (Corning, Manassas, VA, USA, cat. no. 25-058-CI). Accutase was deactivated with an equal volume of the complete culture medium and centrifuged at 200 g for 5 min at RT, resuspended in the culture medium, and plated again in T75 culture flasks at a density of 1 × 10^4^ cells/mL for the experiments.

### 4.2. BMP-2 and FGF-2 Supplementation

To determine the osteogenic stimulatory effect of BMP-2 and FGF-2 on BM-MSCs, the cells were treated with 100 ng/mL of BMP-2 (Stem Cell Technologies, Grenoble, France, cat. no. 78004.1) and/or 20 ng/mL FGF-2 (Merck, Saint Louis, MO, USA, cat. no. F0291). BM-MSCs cultured in the αMEM complete medium were used as a control. The supplemented and control media were changed every three days. The experiments on osteogenic induced BM-MSCs were performed after 7, 14, and 21 days of incubation.

### 4.3. Cell Proliferation and Doubling Time Calculation

The proliferative activity of BM-MSCs cultured with BMP-2 and/or FGF-2 was analyzed using the MTT assay. The cells were seeded in triplicate at a density of 2 × 10^3^ cells/well in 96-well plates. Cells were allowed to attach to the plates for the first 4 h, after which 10 µL of a 4 mg/mL MTT (Merck, Saint Louis, MO, USA, cat. no. M2128) solution was added. After incubation, over the next 4 h at 37 °C, the medium was aspirated and 100 µL of DMSO (POCh, Gliwice, Poland, cat. no. 363550117) was added to solubilize the purple formazan crystals. Optical density was measured at 540 nm with a Wallac Victor2 microplate reader (Perkin Elmer LAS, Waltham, MA, USA). The cell proliferation was analyzed after 24, 48, 72, and 96 h. The medium in each well was replaced after three days. The MTT proliferation assay was repeated three times.

To calculate the cell doubling time (DT), the following formula was used:(1)$$DT=T\cdot \frac{ln2}{ln\frac{N_{T}}{N_{0}}}$$
where *T* is the incubation time (hours), *N_T_* is the number of cells after the incubation time, and *N*_0_ is the number of cells initially harvested. The incubation time point was 48 h, based on the exponential phase. The number of cells after the incubation time was obtained from the growth curve from the MTT assay. The standard curve was plotted to determine the relationship between cell number and absorbance at 540 nm.

### 4.4. Flow Cytometry

BM-MSCs cultured in three different media: αMEM, αMEM + FGF-2, and αMEM + FGF-2 + BMP-2 at different time points of incubation, 7 and 21 days, were analyzed for the basic MSC surface markers using flow cytometry and a FACS Calibur platform (BD Bioscience, Cambridge, UK). Briefly, cells were harvested with the Accutase Cell Detachment Solution (Corning, Manassas, VA, USA), washed in PBS (IIET PAS, Wroclaw, Poland), and resuspended in PBS at a concentration of 2 × 10^6^ cells/mL. For direct flow cytometry, 50 µL of BM-MSCs and an appropriate amount of a conjugated primary antibody (Appendix A) were added to the FACS tubes and incubated for 30 min at 4 °C in the dark. Then, the cells were washed in 1 mL of PBS. After centrifugation, the cells were resuspended in 100 µL of PBS and analyzed. For indirect labelling, 50 µL of BM-MSCs were incubated with the primary antibody at 4 °C for 30 min. Next, the cells were washed and resuspended in a solution of a diluted fluorochrome-labeled secondary antibody in 3% BSA/PBS (Gibco, Carlsbad, CA, USA) and incubated in the dark at 4 °C for 30 min. After incubation, the cells were washed, resuspended in 100 µL PBS, and analyzed using FACS Calibur (Becton Dickinson, San Jose, CA, USA). The data from the flow cytometry experiments were visualized using the Flowing Software version 2.5.1. All used antibodies and their dilutions are listed in Appendix A.

### 4.5. Immunofluorescence Staining

To detect osteogenesis-related proteins, collagen type I, and osteocalcin, the BM-MSCs were cultured in 96-well plates at a density of 1 × 10^3^ cells/well in an αMEM- and αMEM-medium supplemented with BMP-2 and/or FGF-2. The αMEM-medium was used as a control. For immunostaining, performed after 7, 14 and 21 days of incubation, the culture media were aspirated, and the cells were washed with PBS. Next, BM-MSCs were fixed in 100 µL of 3.7% paraformaldehyde for 20 min at RT and washed again in PBS. The cells were then covered with 50 µL of 3% BSA/PBS to block the nonspecific bindings and incubated for 1.5 h at 37 °C. Afterwards, the blocking solution was removed, 50 µL of diluted primary antibodies, rabbit anti-collagen type I, and mouse anti-osteocalcin (Appendix A) were added to the wells for 24 h of incubation at 40 °C. Next, each well was washed with PBS three times, and the cells were incubated with 50 µL of secondary goat anti-rabbit or goat anti-mouse FITC-conjugated solutions (Appendix A) for 30 min in the dark at RT. After incubation with secondary antibodies, the wells were washed three times in PBS. For nuclei staining, DAPI (Vector Labs, Burlingame, CA, USA, cat. no. H-1200) was used for 20 min of incubation in the dark at RT, after which the cells were washed with PBS. Immunofluorescence staining was visualized using an Axio Observer inverted fluorescence microscope (Zeiss, Jena, Germany) equipped with a system for image acquisition and digitalization and analyzed using the Zeiss Zen Blue software.

### 4.6. RT-PCR (Reverse Transcription Polymerase Chain Reaction)

To evaluate the expression of the surface marker CD90, RNA was extracted and purified from the BM-MSCs cultured in αMEM with or without BMP-2 and/or FGF-2 for 7, 14, and 21 days using the NucleoSpin^®^ RNA Kit (Macherey-Nagel, Düren, Germany, cat. no. 740955.50) according to the manufacturer’s instructions. Next, the reverse transcription of 1 µg of total RNA from each sample was performed to prepare cDNA using the RevertAid First Strand cDNA Synthesis Kit (Thermo Fisher, Vilnius, Lithuania, cat. no. K1622). To ensure a good quality of DNA examination, PCR for β-actin was performed on 2% agarose gel with ethidium bromide. The synthesized cDNAs were subjected to PCR using DreamTaq DNA Polymerase (Thermo Fisher, Vilnius, Lithuania, cat. no. EP0702). The PCR primer sequences and reaction parameters are shown in Appendix A. The PCR products were analyzed and visualized with 2% agarose gel electrophoresis using a G:BOX system (Syngene, Cambridge, UK). RT-PCR was normalized by the housekeeping gene GAPDH, and the gel bands were quantified using the ImageJ software (National Institutes of Health, Bethesda, Maryland, USA).

### 4.7. Real-Time qRT-PCR

Extraction and reverse transcription of total RNA from the BM-MSCs treated with 100 ng/mL BMP-2 and/or 20 ng/mL FGF-2 for 7, 14 and 21 days was performed as previously described in the RT-PCR methodology. Real-time PCR was conducted with the Power SYBR Green PCR Master Mix (Life Technologies, Warrington, UK) using the ViiA 7 Real-Time PCR System (Applied Biosystems, Foster City, CA, USA). The reactions were carried out in triplicate with the program running: initial denaturation at 95 °C for 10 min, followed by 40 cycles of denaturation at 95 °C for 15 s, annealing at the Tm (°C) of the primers listed in Appendix A for 1 min, and extension at 72 °C for 40 s. All PCR product quantifications were normalized to the housekeeping gene GAPDH. The relative mRNA expression level was calculated using the 2^-ΔΔCT^ method, where the threshold cycle (CT) from the triplicate runs was averaged and calibrated to GAPDH CT.

### 4.8. Multilineage Differentiation

To examine the multilineage differentiation potential of sheep BM-MSCs, the cells were seeded in a 24-well plate at a density of 1 × 10^4^ cells/well and allowed to attach. After overnight incubation, the culture media were changed to the osteogenic, adipogenic, and chondrogenic differentiation media (PromoCell, Heidelberg, Germany, cat. no. C-28013, C-28012, C-28016) with or without BMP-2 and/or FGF-2 in a volume of 400 µL/well for the induced cells and αMEM for the control cells. The media were refreshed every three days. After 14 days of incubation for chondrogenesis and 21 days for osteogenesis and adipogenesis, the differentiation potential was assessed through visualization with appropriate staining. To perform the staining, the differentiation media were removed, the cells were washed with PBS and fixed for 20 min at RT in a 3.7% formaldehyde (Merck, Saint Louis, MO, USA, cat. no. 104003). Next, the formaldehyde was removed, the cells were washed with PBS again and stained at RT with 200 µL Alizarin Red S for 10 min for osteogenic differentiation, Oil Red O for 15 min for adipogenic differentiation, and Alcian Blue for 40 min for chondrogenic differentiation (Merck, Saint Louis, MO, USA, cat. no. A5533, O0625, A3157).

### 4.9. Sheep Cytokine Array

To evaluate the impact of FGF-2 and BMP-2 on the cytokine secretory profile of sheep BM-MSCs, the cells were cultured in the αMEM medium supplemented with 2 mM L-glutamine and 1% penicillin/streptomycin without FBS and with the addition of: (1) FGF-2 and (2) FGF-2 with BMP-2 for 7, 14 and 21 days. The cells were seeded in T75 cell culture flasks at a density of 1.9 × 10^4^/cm^2^, in the complete αMEM medium first, and incubated for 24 h. Next, the media were changed for αMEM without FBS supplemented with BMP-2 and/or FGF-2. The media were changed every three days. After the time points, the supernatants were collected into 15 mL tubes. To remove cellular debris, the supernatants were centrifuged at 1200 rpm for 10 min at RT, placed into the new tubes, centrifuged once again at 1600 rpm for 10 min at RT, placed into the new tubes and stored in −20 °C. The secretory profile of BM-MSC cytokines was examined using the semi-quantitative C-Series Ovine (Sheep) Cytokine Array C1 Kit (Ray- Bio^®^, Norcross, GA, USA, cat. no. AAO-CYT-1-8) according to the manufacturer’s protocol. Briefly, membranes were placed into the wells of the incubation tray and incubated at RT for 30 min with 2 mL of blocking buffer. Next, the samples were aspirated and 1 mL of supernatants were added for overnight incubation at 4 °C. The next step was to wash the membranes and incubate them with 1 mL of biotinylated antibody cocktail for 2 h at RT. The second wash was performed to remove the unbound antibody, after which 1 mL of HRP-Streptavidin was incubated with the membranes at RT for 2 h. At the end, the membranes were washed and detected with chemiluminescence using an X-ray film. The differences in relative protein expression were measured using the ImageJ and the Protein Array Analyzer plugin. Next, the data were analyzed in Microsoft Excel-based Analysis Software Tool for the Ovine Cytokine Array C1 kit (Ray-Bio^®^, Norcross, GA, USA). The results are presented on heat maps created in GraphPad Prism version 7 (GraphPad Software, Inc., San Diego, California, USA).

### 4.10. Statistical Analysis

All statistical analyses were calculated with the GraphPad Prism version 7. The one-way analysis of variance (one-way ANOVA) with Dunnett’s test for multiple comparison procedures was used to compare the obtained data. *p*-values < 0.05 were considered as statistically significant. All experiments were conducted at least in three independent analyses.

## 5. Conclusions

In this study, we showed that MSCs from sheep bone marrow had a great osteogenic potential when stimulated with FGF-2 and BMP-2. After long-term treatment, the cells still maintained the characteristic phenotype of MSCs and expanded more efficiently compared to the culture in the control medium without cytokines. FGF-2 and BMP-2 enhanced osteogenic differentiation, as confirmed by Alizarin Red S staining and by osteocalcin and collagen type I expression. Furthermore, FGF-2 and BMP-2 upregulated the gene expression of the early osteogenic markers, including *BMP-2*, *osterix*, and *Runx2*, and the late osteogenic markers, *osteocalcin* and *osteopontin*. Our findings demonstrated that sheep BM-MSCs produced a variety of growth factors, cytokines, and chemokines involved in osteogenesis, and supplementation of the culture medium with FGF-2 and BMP-2 affected the secretome profile of the cells.

The improvement of the osteogenic potential of MSCs is very important in research on an innovative treatment for many bone diseases. Co-administration of FGF-2 and BMP-2 seems to be a promising strategy for bone regeneration in in vivo studies on sheep MSCs. Therefore, the next step in this project is to develop a safe cell-based therapy for large bone defects in the sheep model with osteo-inducted BM-MSCs as a preclinical model for cellular therapy in bone regeneration and further potential clinical application.

## Figures and Tables

**Figure 1 ijms-21-09726-f001:**
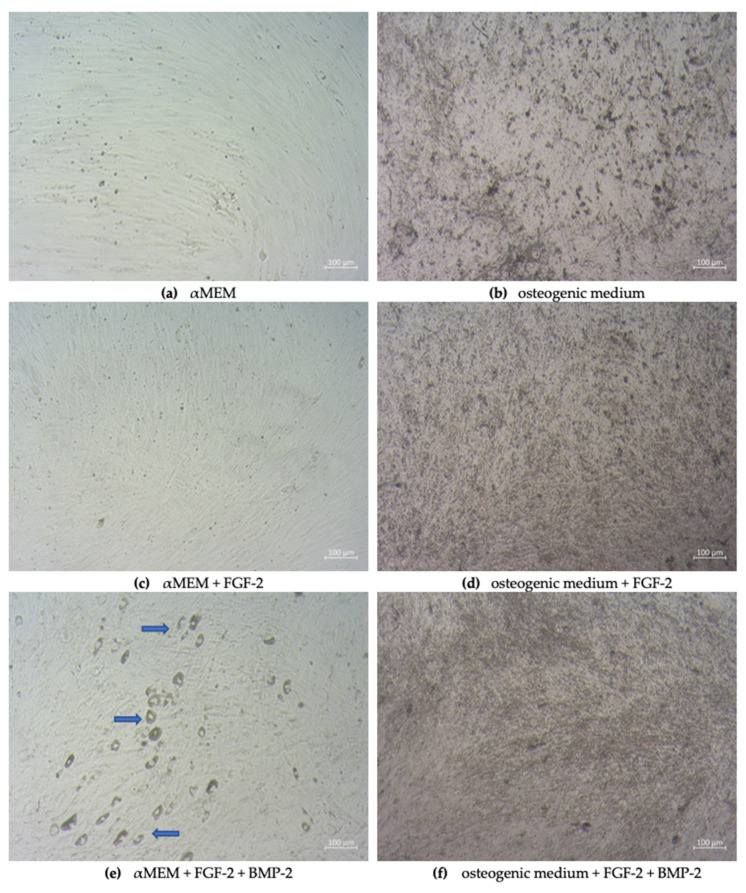
Morphological changes of sheep bone marrow-derived mesenchymal stem cells (BM-MSCs) in different culture conditions. BM-MSCs were cultivated in an αMEM or osteogenic differentiation medium, both supplemented with or without BMP-2 and/or FGF-2 for 21 days. In response to both stimuli (FGF-2 and BMP-2), the cells exhibited different cellular morphology compared to the control culture conditions, i.e., αMEM or osteogenic differentiation medium without any additional cytokines (**a**,**b**). Cells treated with FGF-2 were smaller and grew more densely (**c**). FGF-2 and BMP-2 added to the complete αMEM medium showed an osteogenic differentiation of the cells (**e**), and when added to the osteogenic differentiation medium, they altered the size and density of calcium acceleration (**d**,**f**).

**Figure 2 ijms-21-09726-f002:**
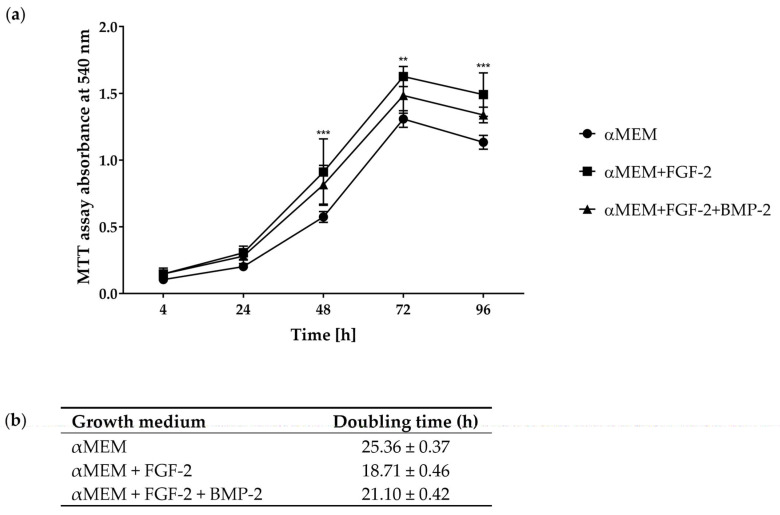
(**a**) Growth curves of sheep BM-MSCs treated with or without BMP-2 and/or FGF-2 assessed with the MTT assay. Compared to the cell control cultured in αMEM, the proliferation of BM-MSCs treated with BMP-2 and/or FGF-2 increased, whereas the proliferation of cells cultured in αMEM supplemented with FGF-2 was the highest. ** *p* ˂ 0.005, *** *p* ˂ 0.0001. (**b**) Doubling time of ovine BM-MSCs cultured in growth media supplemented with or without the cytokines FGF-2 and BMP-2 in three independent experiments performed in triplicate.

**Figure 3 ijms-21-09726-f003:**
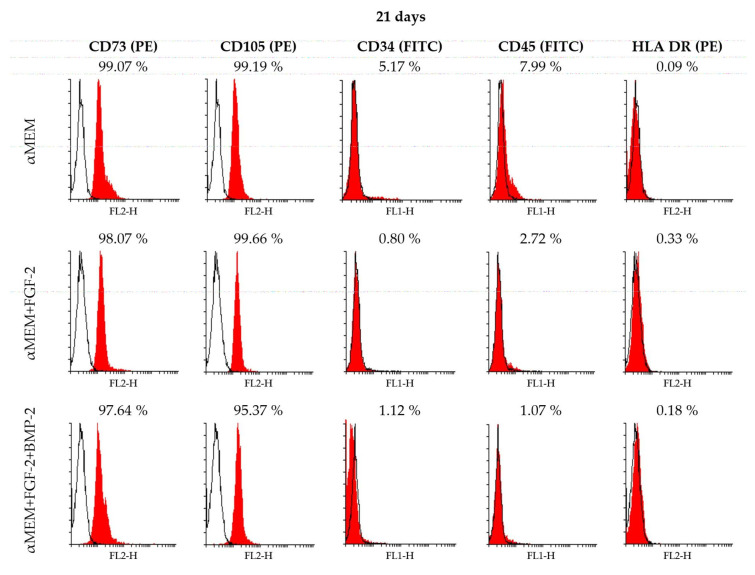
Flow cytometry analysis of BM-MSCs cultured in the control medium αMEM and in a medium supplemented with BMP-2 and/or FGF-2 for 21 days. The expression of the MSC-specific antigens CD73 and CD105 was detected for all cells cultured in different conditions. Cells expressing the hematopoietic markers CD34, CD45, and HLA DR were absent or detected at low levels on day 21 of observation in all culture conditions.

**Figure 4 ijms-21-09726-f004:**
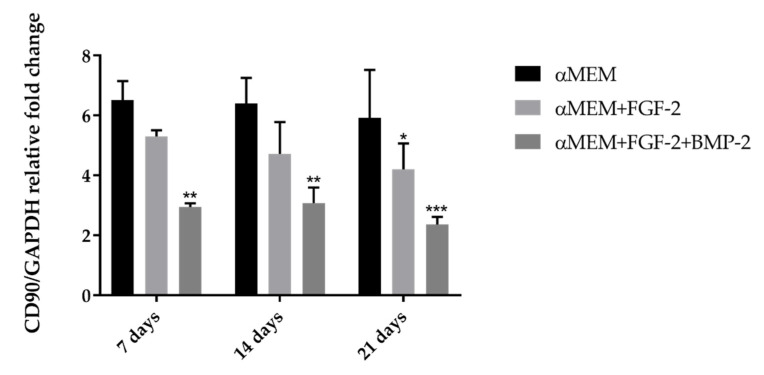
RT-PCR (reverse transcription polymerase chain reaction) analysis for *CD90* gene expression of BM-MSCs cultured in the αMEM medium supplemented with or without BMP-2 and/or FGF-2 for 7, 14, and 21 days. The highest level of *CD90* was observed in cells cultured in the control medium αMEM for all time points. In contrast, BM-MSCs treated with FGF-2 and BMP-2 were characterized with the lowest level of *CD90*. The experiment was assessed three times in duplicate. * *p* ˂ 0.05, ** *p* ˂ 0.005, *** *p* ˂ 0.0001.

**Figure 5 ijms-21-09726-f005:**
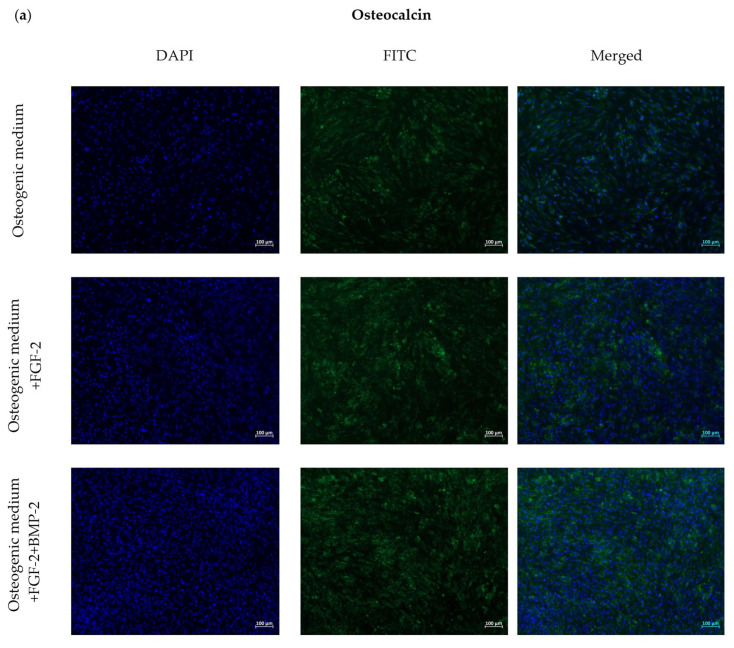
Representative images for the immunofluorescence staining of osteocalcin (Ocl) (**a**) and collagen type I (ColI) (**b**), expressed by BM-MSCs cultured in an osteogenic differentiation medium with or without BMP-2 and/or FGF-2 for 21 days. The Ocl and ColI were stained with FITC in green and nuclei in blue with DAPI. Immunofluorescence staining for osteocalcin (**c**) and collagen type I (**d**) was quantified using the ImageJ software. FGF-2 and BMP-2 promoted the expression of osteogenic-related proteins, which resulted in a moderately more intense fluorescence in the treated cells compared to the control cells cultured in the osteogenic differentiation medium alone.

**Figure 6 ijms-21-09726-f006:**
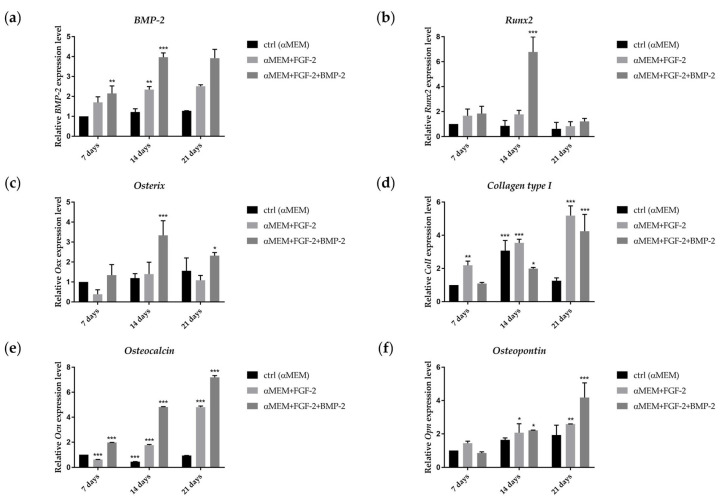
Real-time qRT-PCR analysis for the osteogenic differentiation gene marker of BM-MSCs following treatment with BMP-2 and/or FGF-2 for 7, 14, and 21 days. mRNA for *BMP-2*, *Runx2*, *osterix* (*Osx*), and *collagen type I* (*ColI*) (**a**–**d**) characterizes the early stage of osteogenesis; *osteocalcin* (*Ocn*) and *osteopontin* (*Opn*) (**e**,**f**) characterize the expression in the late stage of differentiation into osteogenic cells. Three different experiments were performed. * *p* ˂ 0.05, ** *p* ˂ 0.005, *** *p* ˂ 0.0001.

**Figure 7 ijms-21-09726-f007:**
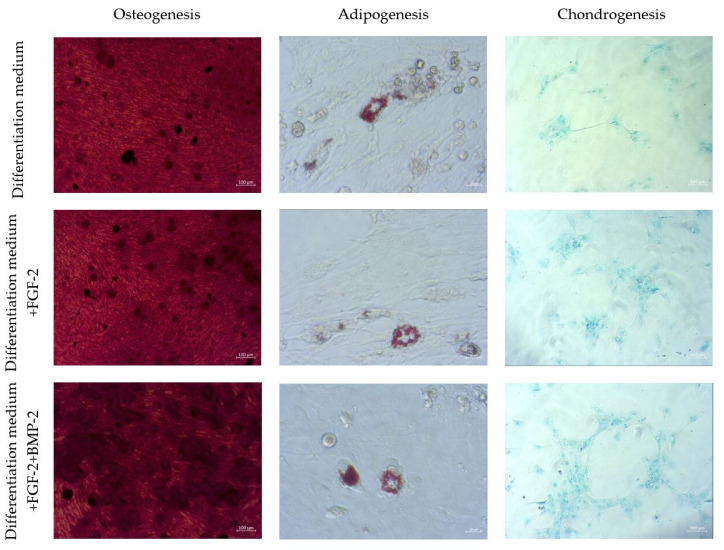
Multilineage differentiation of sheep BM-MSCs. Osteogenic differentiation was assessed using Alizarin Red S staining, adipogenic differentiation using Oil Red O, and chondrogenic differentiation using Alcian Blue staining.

**Figure 8 ijms-21-09726-f008:**
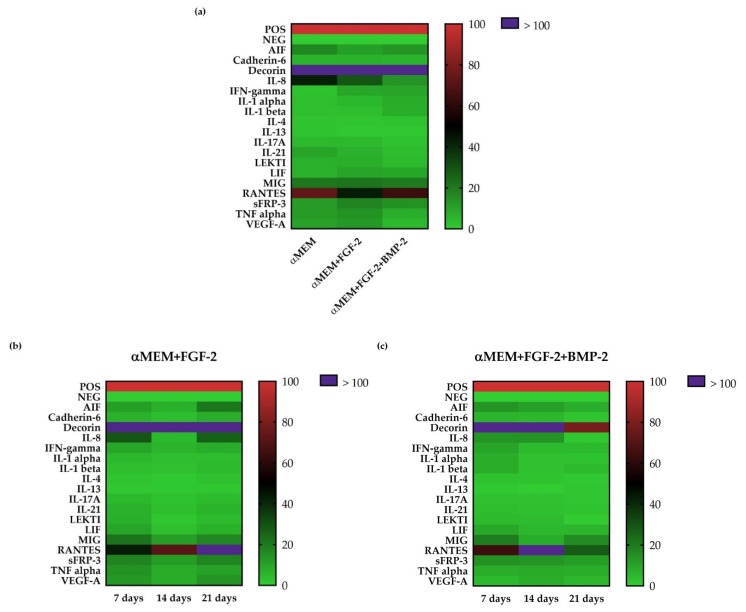
(**a**) Secretion profile of sheep BM-MSCs using the semi-quantitative C-Series Ovine (Sheep) Cytokine Array C1 Kit, depending on culture conditions. (**b**) Secretome of BM-MSCs treated with FGF-2 alone, or (**c**) FGF-2 and BMP-2 after 7, 14 and 21 days of incubation.

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
