# Peer review of "Biological Characteristics and Osteogenic Differentiation of Ovine Bone Marrow Derived Mesenchymal Stem Cells Stimulated with FGF-2 and BMP-2"

_ijms, 2020, doi:10.3390/ijms21249726_

Round 1

Reviewer 1 Report

Review report

This is an interesting article which exploring the bioactive effects of FGF-2, BMP-2 or their combination in promoting the osteogenic differentiation of MSCs.

However, I think there are still some points that needed to be clarified:

Figure 1: I think the figures shown in figure 1 did not provide much information regarding the morphological changes of MSCs under different culture conditions. Perhaps higher resolution images or high power magnification images would be more adequate to display the differences.

Figure 3: I appreciate your extensive efforts (flow cytometric analysis) to demonstrate that the MSCs maintain their differentiation potential throughout 21 days culture period under different treatments. However, wouldn’t it be more important and beneficial for the authors to show osteogenic CD markers of MSCs after osteogenic induction? eg. CD 10 and CD 92 ?

Line 345-347: This phenomenon may be the effect of an ongoing osteogenic differentiation of the examined ovine BM-MSCs, as demonstrated in studies on the osteogenic differentiation of human dental pulp stem cells and a decreased  expression of the MSC surface marker CD90

--- this is indirect evidence, could the author provide more direct evidence showing that the MSCs are undergoing the induced osteogenic differentiation?

Figure 5. Given that the supplement of FGF-2 or its combination with BMP-2 into the osteogenic medium did not show any additional effect on osteogenic gene expression, How would you interpret this data?

Line 314-315: The mineralization of the matrix in the BMP-2- and/or FGF-2-induced cells was higher compared to the 315 cells cultured in an untreated osteogenic differentiation medium. Where is the evidence?

Reviewer 2 Report

Ref# ijms-1020878

Title: Biological characterisitics and osteogenic differentiation of ovine bone marrow derived mesenchymal stem cells stimulated with FGF2 and BMP2

In this manuscript, Gromolak et al. assessed biological changes of the sheep bone marrow MSCs following FGF and BMP2 treatment. Their results showed that FGF and BMP2 have synergistic effect to maintain MSC properties and promote osteogenic differentiation, which has translational significance for bone damage repair. The following comments that need to be paid attention:

  1. The main object of this study is to show that FGF and BMP2 promote osteogenic differentiation of sheep bone marrow mesenchymal cells (BM-MSCs), which is potential therapy for bone damage. If it is the case, the most important part of this research--in vivo studies, which indicate sheep BM-MSCs following FGF and BMP2 treatment could accelerated damaged bone repair, are missing.
  2. The manuscript in current presentation lacks logical thinking. For example, the conclusion in the abstract described as “the results suggest that sheep osteogenic induced BM-MSCs may be used as an effective strategy to study bone repair in the preclinical large animal model”, however, the introduction section talks human aging related bone disorders such as osteoarthritis and osteoporosis, which is definitely unrelated.
  3. The scientific rigorous must be improved. For instance, catalog numbers for major reagents for instance BMP2, FGF2, MTT, NucleoSpin RNA kit etc. are missing. Moreover, to avoid confusion, some terms such as RT-PCR, qRT-PCR should be replaced with their full name. In addition, some unit for instance, line 531 “1mg of total RNA’ is incorrect. In the methods section, the authors indicated CD90 and b-actin was investigated using 2% agarose gel, however, this data is present as real-time PCR for CD90 and GAPDH, not b-actin.
  4. The data presentation must improve. Table 1 and table 2 should move to supplementary. Figures 1, 3, 5 should be clear labeled significant area or specific cell types. It is not well understood that authors listed all immunofluorescence images which seem no significant difference. (a), (b) and (c) in Figure 8 should be combined.
  5. Comprehensive and professional English editing is required.

Reviewer 3 Report

The manuscrip is intersting and well write.The disegne of the study is adequate and results are well presented.

.I think that the manuscript could be published without modifications.

Author Response

The Authors are grateful for Reviewer 3 for recommendation of the manuscript for publication.

Round 2

Reviewer 2 Report

Great appreciate the authors' efforts to take the comments and revise the manuscript! Although the manuscript is much better that previous version, however, honestly, I still have mixed feeling. On one hand, the authors spent lots of efforts on this work, which should be paid off, but on the other hand, the work present as current version is still scientifically immature. The main concern, based on our own research, is some growth factors, including bone morphogenic proteins for instance BMP4 working well in vitro in cell culture system, but they failed in multiple clinical trials for treating human diseases, due to their biological characteristics such as uncomplete cell cycle arrest, short half-life etc. To this end, it is unacceptable for publication without in vivo animal testing.